# Radiotherapy Is Associated with an Accelerated Risk of Carotid Atherosclerosis in Patients with Nasopharyngeal Carcinoma: A Nine-Year Prospective Follow-Up Study

**DOI:** 10.3390/cancers14051234

**Published:** 2022-02-27

**Authors:** Cheng-Hsien Lu, Yun-Ru Lai, Fu-Min Fang, Teng-Yeow Tan, Wen-Chan Chiu, Dong-Yi Hsieh, Chih-Cheng Huang, Chia-Yi Lien, Ben-Chung Cheng, Chih-Yen Chien, Tai-Lin Huang, Chia-Te Kung

**Affiliations:** 1Department of Neurology, Kaohsiung Chang Gung Memorial Hospital, Chang Gung University College of Medicine, 123 Ta Pei Road, Niao Sung District, Kaohsiung City 83301, Taiwan or yunru.lai@gmail.com (Y.-R.L.); b9202095@cgmh.org.tw (D.-Y.H.); hjc2828@gmail.com (C.-C.H.); u9301024@cgmh.org.tw (C.-Y.L.); 2Center for Shockwave Medicine and Tissue Engineering, Kaohsiung Chang Gung Memorial Hospital, Chang Gung University College of Medicine, Kaohsiung City 83301, Taiwan; 3Department of Biological Science, National Sun Yat-sen University, Kaohsiung City 80424, Taiwan; 4Department of Neurology, Xiamen Chang Gung Memorial Hospital, Xiamen 361126, China; 5Department of Radiation Oncology, Kaohsiung Chang Gung Memorial Hospital, Chang Gung University College of Medicine, Kaohsiung City 83301, Taiwan; fang2569@cgmh.org.tw; 6Department of Internal Medicine, Kaohsiung Chang Gung Memorial Hospital, Chang Gung University College of Medicine, Kaohsiung City 83301, Taiwan; testercwt@gmail.com (W.-C.C.); benzmcl@gmail.com (B.-C.C.); victor99@cgmh.org.tw (T.-L.H.); 7Department of Otorhinolaryngology, Kaohsiung Chang Gung Memorial Hospital, Chang Gung University College of Medicine, Kaohsiung City 83301, Taiwan; cychien3965@gmail.com; 8Department of Emergency Medicine, Kaohsiung Chang Gung Memorial Hospital, Chang Gung University College of Medicine, Kaohsiung City 83301, Taiwan; kungchiate@gmail.com

**Keywords:** atherosclerosis, inflammation, nasopharyngeal carcinoma, oxidative stress and radiotherapy, platelet activation

## Abstract

**Simple Summary:**

Radiotherapy has become a common treatment modality for nasopharyngeal carcinoma, and concurrent chemoradiation therapy prolongs the survival of patients. Radiotherapy leads to the increased generation of reactive oxygen species and free radicals to destroy abnormal cell growths within normal cell environmental conditions. However, the late effects on “bystander” organs (e.g., extracranial arteries) among survivors after radiotherapy have become increasingly prevalent. This study showed radiation-related extracranial vasculopathy is a progressive process that begins immediately after radiotherapy with significantly increased carotid IMT compared to the control group during the 9-year follow-up. Chronic inflammation and oxidative stress might serve to drive the process and also contribute to increased platelet activation.

**Abstract:**

Radiation-related extracranial vasculopathy is a common late effect after radiation in patients with nasopharyngeal carcinoma (NPC). We proposed the hypothesis that radiation-related extracranial vasculopathy is a progressive process that can begin immediately after radiotherapy and persist for a longer period, and inflammation and oxidative stress may play a pivotal role in this process. Thirty-six newly diagnosed NPC patients were assessed with B-mode ultrasound for the common carotid artery (CCA) intima media thickness (IMT) measurement as well as surrogate markers at three different stages (baseline, immediately after concurrent chemoradiation therapy (CCRT), and 9 years after enrollment). A healthy control group was also recruited for comparison. Surrogate markers including a lipid profile, HbA1c, inflammation, oxidative stress, and platelet activation markers were assessed. The mean CCA IMT in the NPC group were increased immediately after CCRT (*p* = 0.043). The mean CCA IMT value after a 9-year follow-up also showed a significant increase in NPC and control group, respectively (*p* < 0.0001 and *p* < 0.0001, paired *t* test). The annual increase mean CCA IMT (mm) was 0.053 ± 0.025 and 0.014 ± 0.013 in NPC and control group, respectively (*p* < 0.0001). The baseline high sensitivity CRP (*hs*-CRP), thiol, TBARS, and CD63 level were significantly higher in the NPC group (*hs*-CRP, *p* = 0.001, thiol, *p* < 0.0001, TBARS, *p* = 0.05, and CD63 level, *p* = 0.04). The thiol and TBARS levels were significantly lower in NPC patients immediately after CCRT (thiol, *p* < 0.0001, and TBARS, *p* = 0.043). The CD62P level was significantly higher while the thiol level was significantly lower in the NPC group after a 9-year follow-up (CD62P level, *p* = 0.007; and thiol level, *p* = 0.004). Radiation-related extracranial vasculopathy is a progressive process that begins immediately after radiotherapy with significantly increased carotid IMT compared to the control group during the 9-year follow-up. Chronic inflammation and oxidative stress might serve to drive the process and also contribute to increased platelet activation.

## 1. Introduction

Nasopharyngeal carcinoma (NPC) is characterized by its unique geographic distribution in southeastern China and Taiwan, and both environmental factors and genetic traits contribute to its development [1]. Concurrent chemoradiation therapy (CCRT) prolongs survival in patients with NPC [2,3]. With the increased number of five-year cancer survivors, survivorship has become an important issue [3,4]. A recent Taiwan National Health Insurance Research Database study demonstrated that NPC survivors not only had higher stroke risks but the age of onset of stroke was also 10 years earlier than that for the general population [3].

Radiation therapy leads to the increased generation of reactive oxygen species (ROS) and free radicals to destroy abnormal cell growths within normal cell environmental conditions. When cellular repair and free radical scavenger systems cannot counteract these insults, oxidative damage occurs and finally has the consequence of abnormal cellular structure and function [3]. The injured endothelium can promote inflammation through the up-regulation of the cells’ adhesion molecules by binding to circulating leukocytes as an important step during atherosclerosis [5,6,7]. There is accumulating evidence indicating that inflammation and oxidative stress may play a pivotal role in the process of radiation-related extracranial vasculopathy [8,9].

In clinical practice, radiation injury is believed to accelerate atherosclerosis in the arteries in a progressive process without delineating the acute from the late radiation effects [10,11]. These radiation effects can be seen several months after radiation and may last for years [10]. For research on radiation-related vasculopathy in NPC, most previous studies are retrospective with variable less strict selection [12], and included patients with different types of heads and neck cancer [12,13,14], variable follow-up periods with cross-sectional study [13,15,16,17], national database epidemiological study [3], and a systematic review and meta-analysis [12,18].

Carotid intima media thickness (IMT) can serve as an indicator not only in the severity of atherosclerotic disease [19] but also in the early stage of atherosclerosis [20]. The findings of the earliest post-irradiation effect in asymptomatic patients by Doppler sonography study is an increase in carotid IMT [15]. In contrast, the existence of carotid plaques demonstrates a higher cerebrovascular risk than an increase in the carotid IMT [21].

To our knowledge, little research is known about the serial changes of carotid IMT and biomarkers in inflammation and oxidative stress and platelet activation in both the acute phase and long-term follow-up in patients with NPC before and after radiotherapy. This study tested the hypothesis that radiation-related extracranial vasculopathy is a progressive process that can begin immediately after radiotherapy and persist for a longer period, and inflammation and oxidative stress may play a pivotal role in this process. We hope the successful translation of these approaches can offer the promise of long-term survivorship in patients of NPC.

## 2. Materials and Methods

### 2.1. Study Design and Patient Selection

This prospective, case-control study was conducted at the main referral hospital and a tertiary medical center in southern Taiwan. We evaluated newly diagnosed NPC patients from September 2010 to March 2012. The exclusion criteria were (1) there was evidence of head and neck cancer with relapse and (2) a history of cerebral infarctions with extracranial vessels atherosclerotic narrowing (e.g., heavy calcification and multiple plaques). Finally, only 36 cases were enrolled in this study. All participants received verbal and written information for the purpose, and they signed informed consent. For clinical comparison, 36 aged- and sex-match healthy volunteers formed the control group.

### 2.2. Diagnostic Criteria and Therapeutic Regimens

The multi-disciplinary teams administered treatment for all patients after pathologists histologically diagnosed NPC. According to the American Joint Committee on Cancer (AJCC) staging system for NPC, the distribution of the 36 patients was as follows: two Stage I, eight Stage II, 11 Stage III, and 15 Stage IV. The prescribed dose ranged from 6640 to 7560 cGy depending on the tumor stage. The technical details of radiotherapy and CCRT chemotherapy regimen for NPC have been described previously [15].

### 2.3. Clinical Assessment

Complete medical history was recorded, including age at disease onset, sex, height and weight, body mass index (BMI), AJCC staging system for NPC, underlying disease and vascular risk factors, mean radiation dose delivered to either side of CCA IMT, and follow-up period after radiotherapy. All patients received complete medical and neurologic examinations, peripheral blood study and extra-cranial color-coded duplex sonography (ECCS).

### 2.4. Assessment of Atherosclerosis

Images were obtained from a B-mode ultrasound system (Philips HDI 5000 System, ATL-Philips, Bothell, WA, USA) equipped with a 4–10 MHz linear array transducer. The obtained images were transferred to a workstation and the IMT was automatically measured using a computer software program (Q-LAB, ATL-Philips). The plaque was defined as a localized wall thickening at least twice the thickness of the adjacent IMT [15]. The standardized protocol was previously described [15]. One neurologist (T.-Y.T.) performed the measurements of mean CCA IMT by quantifying software in a single-blind fashion blind to the patients and control subjects. In this study, we further defined the annual increase mean CCA IMT (mm) and the formula was as follows: [(Mean CCA IMT (mm) at end-point of follow-up-mean CCA IMT (mm) at enrollment)/duration between two examinations (months)] × 12 (months)

### 2.5. Biochemical Analysis

Blood samples were collected into Vacutainer SST tubes (BD, Franklin Lakes, NJ, USA) by venipuncture. The blood was allowed to clot at room temperature for a minimum of 30 min. All samples were collected after centrifugation at 3000 rpm for 10 min at 4 °C, isolated, and immediately stored in multiple aliquots at −80 °C. The samples of biochemical data and peripheral blood study including white blood cell (WBC), red blood cell (RBC) and platelet counts, triglycerides, total, high-density lipoprotein (HDL), and low-density lipoprotein (LDL) cholesterol, blood glucose, glycosylated hemoglobin (HbA1c), and high sensitive C-reactive protein (hs-CRP) were analyzed by the central laboratory of our hospital.

### 2.6. Biomarkers for Oxidative Stress and Anti-Oxidative Capacity

We evaluated the oxidative stress condition by measuring the serum thiobarbituric acid-reactive substance (TBARS) while the anti-oxidative capacity in response to the increased oxidative damage was evaluated by measuring the serum level of total reduced thiols. Serum TBARS levels were measured using a well-established method for detecting lipid peroxidation with a commercially available assay kit (Cayman Chemical, Ann Arbor, MI, USA, cat. no. 10009055) according to the manufacturer’s instructions [22].

Flow cytometry was done according to the modified protocol for flow cytometric characterization of platelet function as previously described [23]. The platelet activation marker was determined by CD62P, CD63, and CD40L. Platelet-bound anti-CD62P, anti-CD63, and CD40L antibodies were determined by analyzing 10,000 platelets for PE positive fluorescence. Results are expressed as a percentage of antibody-positive platelets.

### 2.7. Statistical Analysis

Continuous data were expressed as mean ± SD or median (interquartile range [IQR]). Continuous variables were compared using the student’s *t*-test and the categorical variables were compared using the chi-squared or Fisher’s exact tests. The parameters of the carotid IMT, biochemical data, and biomarkers of oxidative stress and platelet activation markers at baseline (Before CCRT) and three months (After CCRT) and 9-years after enrollment in the patient group were compared using a paired *t*-test. The parameters of the carotid IMT and biochemical data at baseline and 9-years after enrollment in the control group were compared using a paired *t*-test. The repeated-measures ANOVA was used to compare the parameters of carotid IMT and biochemical data at two different stages of the study (baseline and 9 years after enrollment), with sex and age as potential confounding variables. All statistical analyses were conducted using the Statistical software (IBM SPSS Statistics v23, IBM, Redmond, WA, USA).

## 3. Results

### 3.1. Baseline Characteristics of the Study Patients

The baseline characteristics of the 36 adult NPC patients and 36 healthy volunteers are listed in Table 1. The age at enrollment, male to female gender difference, body weight, and BMI was similar between two groups. The mean diastolic blood pressure during ECCS examination was higher in the NPC group (*p* = 0.001). The mean duration between the date of the complete course of radiotherapy and the last follow-up (months) in the NPC group was 105.1 ± 7.0 months. One patient received carotid stenting for carotid critical stenosis, though none had symptomatic cerebral infarction. In total, 13 cases died with the overall fatality being 36% (13/36) during the 9-year follow-up period.

### 3.2. Serial Changes of Carotid IMT and Plaques between NPC and Control Groups

The serial changes of carotid IMT values and plaques between NPC and control groups at baseline and after 9-year follow-up are listed in Table 2. The baseline IMT of CCA values on either side and the mean CCA IMT values were also recorded in the NPC group immediately after CCRT. The baseline CCA IMT values on either side and the mean CCA IMT values were similar but slightly higher in the control group (left CCA IMT, *p* = 0.44; right CCA IMT, *p* = 0.36; and mean CCA IMT (average), *p* = 0.37). Carotid plaque was detected in 5 NPC patients (5/36, 13.8%) but only two (2/36, 5.5%) in the control group. The CCA IMT values on either side and the mean CCA IMT values in the NPC group increased immediately after CCRT (left CCA IMT, *p* = 0.042; right CCA IMT, *p* = 0.06; and mean CCA IMT (average), *p* = 0.043).

The CCA IMT values on either side and the mean CCA IMT values between baseline and 9-year follow-up in the NPC group showed a significant increase (left CCA IMT, *p* < 0.0001; right CCA IMT, *p* < 0.0001; and mean CCA IMT (average), *p* < 0.0001). The CCA IMT values on either side and the mean CCA IMT between baseline and 9-year follow-up in the control group also showed a significant increase (left CCA IMT, *p* = 0.001; right CCA IMT, *p* = 0.001; and mean CCA IMT (average), *p* < 0.0001). The CCA IMT values on either side and the mean CCA IMT after a 9-year follow-up between NPC and control groups also showed a significant increase in NPC group (left CCA IMT, *p* = 0.001; right CCA IMT, *p* = 0.009; and mean CCA IMT (average), *p* = 0.001). The annual increase means CCA IMT (mm) was 0.053 ± 0.025 and 0.014 ± 0.013 in the NPC and control group, respectively (*p* < 0.0001). However, the NPC and control group at two time periods (baseline and 9-years follow-up) by mean of repeated measure ANOVA after controlling age and sex did not show statistical significance (left CCA IMT, *p* = 0.18; right CCA IMT, *p* = 0.22; and CCA IMT (average), *p* = 0.17) (Figure 1).

### 3.3. Assessment of the Relationship between the Radiation Dose and the CCA IMT Change

The mean radiation dose on the right and left CCA were 6769.8 ± 510.1 and 6788.7 ± 488.7 cGy, respectively. The CCA IMT changes are defined as the value of CCA IMT at the 9-year follow-up minus the value of CCA IMT at baseline. Correlation analysis was explored to evaluate the relationship between the mean radiation dose delivered to either side of CCA and the ipsilateral side CCA IMT change. The results were as follows: right CCA IMT change and mean radiation dose (r = 0.38, *p* = 0.15) and left CCA IMT change and mean radiation dose (r = 0.33, *p* = 0.21).

### 3.4. Serial Changes of Biochemical Data, Oxidative Stress, and Platelet Activation Markers between NPC and Control Groups

Serial changes of biochemical data, oxidative stress, and platelet activation markers between NPC and control groups are listed in Table 3 and Figure 2a–c and Figure 3a,b. Regarding the baseline biochemical data, oxidative stress, and platelet activation markers between NPC and control groups, there were no significant differences between the two groups in WBC counts, platelet counts, LDL, total cholesterol, LDL, CD40L and CD62P level. However, serum triglyceride, hs-CRP, blood glucose, HbA1c, thiol, TBARS, and CD63 level were significantly higher in the NPC group than in the control group (triglyceride, *p* = 0.001, hs-CRP, *p* = 0.001, glucose, *p* = 0.04, HbA1c, *p* = 0.004, thiol, *p* < 0.0001, TBARS, *p* = 0.05, and CD63 level *p* = 0.04). However, the serum HDL and RBC count was significantly lower in the NPC group than in the control group (*p* = 0.004 and *p* = 0.03).

Regarding the biochemical data, oxidative stress and platelet activation markers in the NPC group before and after CCRT in the NPC group, WBC and RBC count, platelet counts, HbA1c, thiol, and TBARS level were significantly lower in the NPC group (*p* < 0.05) after CCRT (HbA1c, *p* = 0.01, WBC counts, *p* < 0.0001, RBC counts, *p* < 0.0001, platelet counts, *p* < 0.0001, thiol, *p* < 0.0001, and TBARS, *p* = 0.043). However, serum HDL and LDL levels were significantly higher in the NPC group after CCRT (HDL level, *p* = 0.01, and LDL level, *p* = 0.03).

Regarding the inflammation and oxidative stress in NPC cases at baseline before CCRT, the levels of hs-CRP, thiol, and TBARS were 4.4 ± 2.4, 1.6 ± 0.4, and 14.2 ± 3.8 in those patients who survived while those data were 3.1 ± 1.8, 1.2 ± 0.4, and 15.2 ± 5.0 in those patients who died (independent-*t* test, hs-CRP, *p* = 0.63, thiol *p* = 0.009, and TBARS *p* = 0.55). The levels of hs-CRP, thiol, and TBARS were 4.4 ± 2.4, 1.6 ± 0.4 and 14.2 ± 3.8 before CCRT and 1.0 ± 0.4, 1.3 ± 0.3 and 13.0 ± 5.1 after CCRT in those patients who survived (paired-*t* test, hs-CRP, *p* = 0.08, thiol, *p* < 0.0001, and TBARS *p* = 0.47). The levels of hs-CRP, thiol and TBARS were 3.1 ± 1.8, 1.2 ± 0.4 and 15.2 ± 5.0 before CCRT, and 0.7 ± 0.5, 0.8 ± 0.4 and 11.1 ± 3.0 after CCRT in those patients who died (paired-*t* test, hs-CRP, *p* = 0.5, thiol *p* = 0.06, and TBARS *p* = 0.07).

The CD62P level was significantly higher while the thiol level was significantly lower in NPC patients after a 9-year follow-up (CD62P level, *p* = 0.007; and thiol level, *p* = 0.004). The fasting glucose was significantly higher while HDL was significantly lower in the control group after a 9-year follow-up (fasting glucose, *p* < 0.0001, and HDL, *p* < 0.0001). The biochemical data between NPC and control groups after a 9-year follow-up showed that only hs-CRP is significantly higher in the NPC group (hs-CRP, *p* = 0.006).

## 4. Discussion

### 4.1. Major Findings of Our Study

To date only a few clinical researchers have assessed atherosclerotic changes including carotid IMT and plaque in the vessel wall by using ECCS and quantifying software in post-radiotherapy NPC patients [15,16,17]. To our knowledge, this is the first study to assess the CCA IMT in patients with NPC before and immediately after CCRT as well as follow-up for nearly one decade. Our study confirms the posited hypothesis that radiation-related extracranial vasculopathy is a progressive process that begins immediately after radiotherapy with significantly increased carotid IMT compared to the control group during the 9-year follow-up. Regarding the dose–response relationship, we found the relationship between IMT change and the mean radiation dose delivered did not show statistical significance. Since all patients receive high radiation doses delivered to either CCA, it is not surprising that the correlation analysis did not show statistical significance in NPC patients. In contrast, the IMT change showed a significant increase as compared with the control group. Although a causal link between chronic inflammation and oxidative stress and radiation-related extracranial vasculopathy remains to be established, it might serve to drive the pathogenesis of carotid IMT progression and plaque formation as well as increased platelet activation.

### 4.2. Prevalence and Locations of Radiation-Related Vasculopathy

Most clinical studies have focused on the prevalence rate and locations of radiation-related extracranial vasculopathy in NPC with a cross-sectional study with various follow-up periods after radiotherapy [15,24,25,26], a discussion under the topic of head and neck cancer [13,27], or the national health insurance database [3], systemic review and meta-analysis [18]. The prevalence rate of radiation-related cerebral vasculopathy was found to be from 17% to 93% in different studies [13,15,25,26,27,28,29].

Regarding the locations of radiation-related extracranial vasculopathy, its distribution and the extent of arterial involvement may also be different according to the strategy of radiotherapy (e.g., tumor locations and sizes, and staging of tumors). It can lead to lengthy stenosis with the extension of diseased segments well beyond the common carotid bifurcation [30]. These studies showed that internal carotid arteries (ICA) were most commonly involved, followed by CCA, CCA/ICA, external carotid (ECA) and vertebral arteries (VA) [25,26,31]. Although our previous study showed increased CCA IMT and presence of plaque in 36.2% (38/105) with bilateral CCA (84%, 32/38) involvement predominant in NPC patients [15], we only assessed carotid IMT progression and plaque formation in the present study because of the small sample size.

### 4.3. The Role of Inflammation, Oxidative Stress and Platelet Activation in Patients with NPC

Oxidative stress and antioxidant capacity play a crucial role in the process of malignant transformation [32]. On the other hand, growing evidence highlights the role that inflammation and oxidative stress can also play by different metabolic pathways to eliminate cancer cells [33]. Therefore, these biomarkers can be considered useful markers involved in cancer diagnosis and outcome prediction [32,34]. In our study, there was a significantly increased level of oxidative stress and antioxidant capacity at baseline in patients as compared with healthy controls; both the antioxidants and oxidative stress levels tended to decrease immediately after CCRT. In contrast, the antioxidant capacity remains low while oxidative stress levels tend to be higher after a 9-year follow-up period in NPC patients. Chronic inflammation is well-known to predispose to the development and progression of cancer, sustained tissue damage, and tissue repair [35]. Our study demonstrated that the hs-CRP level is higher than the healthy control, tended to decrease immediately after CCRT, and sustained a higher level after a 9-year follow-up period in NPC patients. Platelet activation plays an important role in the pathophysiology of ischemic stroke [36]; increased platelet activation markers (e.g., CD62P, CD63, and CD40L) occurred in both the acute and chronic phase after ischemic stroke [37,38]. Our previous study also showed that patients with large-vessel cerebral infarction elicit higher platelet activation compared to small-vessel infarction [36]. Therefore, platelet activation markers can serve as reliable markers for increased risk of ischemic stroke in those patients with radiation-related extracranial vasculopathy. The CD62P level in our study was significantly higher in NPC patients after a 9-year follow-up (*p* = 0.007).

### 4.4. Risk of Cerebrovascular Events

Late effects on “bystander” organs (e.g., extracranial arteries) among survivors of the NPC patients after radiotherapy have become increasingly prevalent, with secondary increasing cerebrovascular events [12,13,14,39]. Several large epidemiological studies showed an increased ischemic stroke or transient ischemic attack rate after irradiation for head and neck cancer. Most of these studies were national database epidemiological studies [3], and systematic review and meta-analyses [12,18], and had variable follow-up periods with a cross-sectional study [13,15,16,17], or were a discussion under the topic of head and neck cancer [13,27]. However, big data epidemiological studies cannot really assess the discrepancy between radiation-related extracranial vasculopathy and symptomatic cerebral infarction and reflect the real world of NPC survivorship. A multicenter, prospective, hospital-based epidemiological study may provide accurate information about the effect of the radiotherapy dose–response relationship on radiation-related extracranial vasculopathy and clinical symptomatic stroke.

### 4.5. Management of Radiation-Related Extracranial Vasculopathy

Therapeutic guidelines for the prevention of cerebral infarction or transient ischemic attack in extracranial carotid disease recommend optimal medical therapy, including antiplatelet therapy, statin therapy, and risk factor modification [40], while interventional approaches including carotid endarterectomy, carotid angioplasty, and stenting are recommended to those patients with symptomatic extracranial carotid stenosis (>70%) [18]. However, no clinical trial to date has adequately assessed the medical treatment options in primary or secondary stroke prevention in this selected group of patients. The effect of antiplatelet, anticoagulant, antihypertensive, or lipid-lowering therapy in limiting disease progression is therefore unclear and highlights the need for more clinical studies to develop additional drugs.

### 4.6. Study Limitation

Our study had several limitations. First, several underlying diseases and vascular risk factors (e.g., hypertension, DM and hyperlipidemia) and drugs (e.g., statins) could influence the carotid IMT progression, and the annual increase in carotid IMT in each patient could be different. Besides radiotherapy, the only patient who received carotid stenting for critical carotid stenosis was a heavy smoker but without other common vascular risk factors (e.g., hypertension, diabetes, and hyperlipidemia). Cigarette smoking is a well-known vascular risk factor for atherosclerosis. Although one meta-analysis showed a nonlinear dose–response relationship between the number of cigarettes smoked per day and the risk of cardiovascular events [41], the other study showed smoking is associated with dose-related increase of carotid IMT [42]. Second, radiation-related extracranial vasculopathy is a progressive process. Although most of our patients received annual ECCS examination, the timing of the carotid IMT increase in accelerated phases in those patients who had critical stenosis remains undetermined. Third, this study proceeded and followed up ECCS in the NPC group for nearly one decade. The healthy control group only received an annual ECCS study but biomarkers for oxidative stress, antioxidative capacity, and platelet function were only obtained at baseline. Finally, although chronic inflammation and oxidative stress might be involved in the process, as well as increased platelet activation, a causal link between chronic oxidative stress and radiation-related extracranial vasculopathy remains to be established.

## 5. Conclusions

Based on our results, radiation-related extracranial vasculopathy is a progressive process that begins immediately after radiotherapy with significantly increased carotid IMT than the control group during the 9-year follow-up. Chronic inflammation and oxidative stress might serve to drive the process and also contribute to increased platelet activation. ECCS may be added in annual screenings in NPC patients to early detection of radiation-related extracranial vasculopathy and also to assess the discrepancy between radiation-related extracranial vasculopathy and symptomatic cerebral infarction. Although no clinical trial to date has adequately assessed the medical treatment options in primary or secondary prevention in radiation-related extracranial vasculopathy, optimal medical therapy, including antiplatelet therapy, statin therapy, and risk factor modification could be effective.

## Figures and Tables

**Figure 1 cancers-14-01234-f001:**
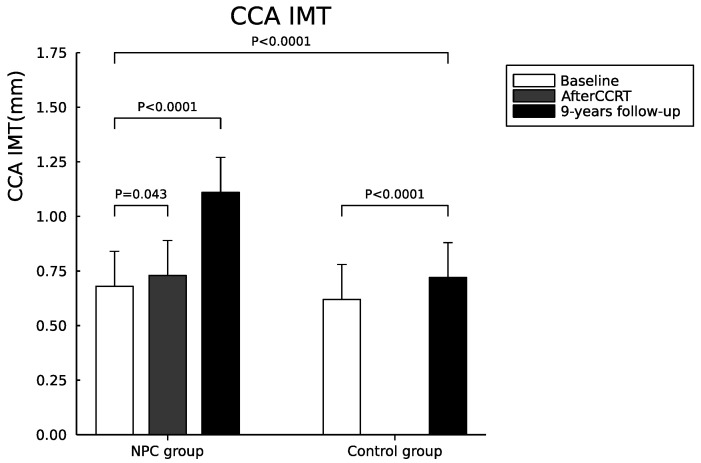
Serial changes of mean CCA IMT in NPC and control groups during the study period.

**Figure 2 cancers-14-01234-f002:**
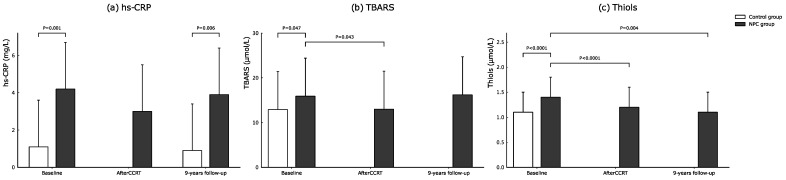
Serial changes of biomarkers of inflammation and oxidative stress including mean hs-CRP (**a**), TBARS (**b**), and Thiols (**c**) in NPC and control groups during the study period.

**Figure 3 cancers-14-01234-f003:**
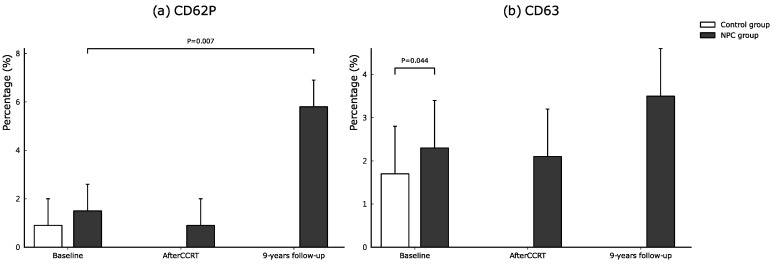
Serial changes of platelet activation markers including CD63 (%) (**a**), and CD62P (%) (**b**) in NPC and control groups during the study period.

**Table 1 cancers-14-01234-t001:** Baseline demographic data in nasopharyngeal carcinoma patients and normal controls.

	NPC Patients (*n* = 36)	Controls (*n* = 36)	*p* Value
Age (years) at enrollment	55.4 ± 12.3	55.3 ± 14.1	0.97
Gender (female/male)	12/24	12/24	1.0
Body mass index, kg/m^2^	24.8 ± 3.3	24.5 ± 2.8	0.2
Height (m)	1.63 ± 0.07	1.63 ± 0.09	0.82
Body weight (Kg)	66.2 ± 11.1	64.6 ± 11.3	0.57
Mean duration between the date of the complete course of radiotherapy and the last follow-up (months)	105.1 ± 7.0	-	
Systolic blood pressure	126.9 ± s12.4	122.2 ± 17.9	0.2
Diastolic blood pressure	77.1 ± 8.0	70.0 ± 9.3	0.001
Smoking (%)	18	2	0.001 *
Staging system of AJCC for NPC			
Stage 1	2	--	
Stage 2	8	--	
Stage 3	11	--	
Stage 4	15	--	

Values are expressed in mean ± SD unless otherwise indicated. Abbreviations: IQR, inter-quartile range; HBA1c, glycosylated hemoglobin. * = Indicated *p* < 0.05.

**Table 2 cancers-14-01234-t002:** Serial changes of carotid IMT and plaques between patients and controls.

	Baseline (before and after CCRT)			Nine-Years Follow-Up	
	Controls (*n* = 36)	NPC Patients (*n* = 36)		Controls ^γ,^^Φ^ (*n* = 36)	NPC Patients ^δ^ (*n* = 23) ^α^
		Before CCRT	After CCRT ^β^		
Carotid IMT and plaque score ^α^					
Right CCA IMT, mm	0.69 ± 0.20	0.65 ± 0.17	0.71 ± 0.23 ^‡^	0.81 ± 0.27 ^#,‖^	1.05 ± 0.26 ^¶^
Left CCA IMT, mm	0.75 ± 0.27	0.71 ± 0.21	0.74 ± 0.22 ^‡^	0.86 ± 0.27 ^#,‖^	1.18 ± 0.27 ^¶^
Mean CCA IMT, mm	0.72 ± 0.22	0.68 ± 0.16	0.73 ± 0.21 ^‡^	0.83 ± 0.25 ^#,‖^	1.11 ± 0.23 ^¶^
Carotid Plaque (%)	2 (5.5%)	5 (13.8%)	5 (13.8%)	5 (13.8%)	11 (47.8%)
Annual increase mean CCA IMT, mm	-	--	--	0.014 ± 0.013	0.053 ± 0.025 ^§^

Values are expressed in mean ± SD unless otherwise indicated. Abbreviations: IMT = intima media thickness; CCA = common carotid artery. ^α^ = 13 cases died by the end-point of follow-up and only 23 cases enrolled for analysis. ^β^ = Before and after CCRT in NPC group were compared by paired *t* test. ^‡^ = Indicated *p* < 0.05; ^γ^ = Baseline and 9-year follow-up in control group were compared by paired *t* test. ^#^ = Indicated *p* < 0.05; ^δ^ = Baseline and 9-year follow-up in NPC group were compared by paired *t* test. ^¶^ = Indicated *p* < 0.0001; ^Φ^ = Nine-year follow-up between NPC and control groups were compared by mean of independent-*t* test, ^‖^ = Indicates *p* < 0.01. ^§^ = Indicates *p* < 0.0001.

**Table 3 cancers-14-01234-t003:** Serial changes of biochemical data, oxidative stress and platelet activation markers between patients and controls.

	Baseline (before and after CCRT)			Nine-Years Follow-Up	
	Controls ^β^ (*n* = 36)	NPC Patients (*n* = 36)		Controls ^ζ,^^Φ^ (*n* = 36)	NPC Patients ^δ^ (*n* = 23) ^α^
		Before CCRT	After CCRT ^γ^		
Biochemical data					
Cholesterol, mg/dL					
Total	184.7 ± 31.5	189.3 ± 39.6	202.7 ± 32.7	180.7 ± 35.3	167.4 ± 41.3
HDL-C	67.9 ± 13.7 *	51.6 ± 14.2	57.5 ± 14.0	53.6 ± 11.8 ^§^	47.3 ± 16.4
LDL-C	98.9 ± 30.6	108.9 ± 29.8	121.5 ± 28.8	98.5 ± 18.0	93.5 ± 34.6
Triglyceride, mg/dL	89.7 ± 45.9 *	137.8 ± 70.0	120.3 ± 50.0	100.4 ± 45.3	118.9 ± 42.8
Glucose, mg/dL	89.7 ± 10.2 *	117.3 ± 75.6	103.0 ± 34.3	105.3 ± 17.5 ^§^	118.9 ± 32.8
HBA1c	5.6 ± 0.3 *	6.2 ± 1.2	5.7 ± 0.6 ^#^	5.9 ± 0.7	6.3 ± 0.8
hs-CRP, mg/L	1.1 ± 0.9 *	4.2 ± 2.5	3.0 ± 2.6	0.9 ± 0.5 *	3.9 ± 3.3
WBC counts (×10^3^/mL)	6.0 ± 1.4	6.3 ± 2.0	4.1 ± 1.3 ^#^	6.3 ± 1.9	7.0 ± 2.9
RBC counts (×10^6^/mL)	5.0 ± 0.7 *	4.6 ± 0.7	3.9 ± 0.6 ^#^	4.5 ± 0.7	4.5 ± 0.7
Platelet counts (×10^3^/mL)	217.6 ± 60.3	223.0 ± 67.1	221.3 ± 70.8 ^#^	248.5 ± 131.9	235.9 ± 60.4
Biomarkers for oxidative stress					
TBARS, μmol/L	12.9 ± 2.4 *	15.9 ± 8.5	13.0 ± 5.3 ^#^	-	16.2 ± 7.9
Thiols, μmol/L	1.1 ± 0.2 *	1.4 ± 0.4	1.2 ± 0.3 ^#^	-	1.1 ± 0.5 ^¶^
Platelet activation markers					
CD40 (%)	0.4 ± 0.2	0.5 ± 0.3	0.6 ± 0.3	-	0.5 ± 0.3
CD63 (%)	1.7 ± 1.1 *	2.3 ± 1.1	2.1 ± 1.1	-	3.5 ± 2.3
CD62P (%)	1.0 ± 0.7	1.5 ± 1.1	0.9 ± 0.6	-	5.8 ± 4.9 ^¶^

Values are expressed in mean ± SD unless otherwise indicated. Abbreviations: IMT = intima media thickness; CCA = common carotid artery. ^α^ = 13 cases died during the end-point of follow-up and only 23 cases enrolled for analysis. ^β^ = Baseline data between NPC and control groups were compared by mean of independent-*t* test, * = Indicated *p* < 0.05. ^γ^ = Before and after CCRT in NPC group were compared by paired *t* test. ^#^ = Indicated *p* < 0.05; ^δ^ = Before CCRT and 9-year follow-up in the NPC group were compared by paired *t* test. ^¶^ = Indicated *p* < 0.05; ^ζ^ = Baseline and 9-year follow-up in control group were compared by paired *t* test. ^§^ = Indicated *p* < 0.05; ^Φ^ = Nine-years follow-up between NPC and control groups were compared by mean of independent-*t* test, * = Indicates *p* < 0.05.

## Data Availability

The datasets used and/or analyzed during the current study are available from the corresponding author upon reasonable request.

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
