# Peer review of "Radiotherapy Is Associated with an Accelerated Risk of Carotid Atherosclerosis in Patients with Nasopharyngeal Carcinoma: A Nine-Year Prospective Follow-Up Study"

_cancers, 2022, doi:10.3390/cancers14051234_

Round 1

Reviewer 1 Report

The authors focus on the risk of developing carotid atherosclerosis after chemoradiotherapy for nasopharyngeal carcinoma within a prospective case-control study. It’s a 4-fold risk? This is a highly interesting topic, especially with the young-age incidence of NPC and the possible curative aim of treatment.  Unfortunately the patient number is limited (n=36) and data were compared to an even smaller ‘age-matched’ control group (n=22). The demographic data in table 1 describe contrarily a non-balanced comparison. 39% of NPC patients had hypertension, 31% had diabetes and 50% were smokers, while only one risk factor (smoker) was seen in 5% of the control group. 13,8% of NPC patients suffered already from carotid plaque before CCRT. Considering the term of having an age-matched control, those with NPC were in mean 4 years older. Why only 22 people within the control group? Why not a matched pairs analysis (1:1, 1:2) based on the known risk factors for developing atherosclerosis? One NPC patient had carotid stenting. Vascular risk factors? We should be really aware about a higher incidence of CCA atherosclerosis in NPC patients after radiotherapy  and clearly need more data for risk assessment. Atherosclerosis is a multifactorial disease, mainly driven by inflammatory reactions. It’s the same for radiation induced atherosclerosis, or predominantly a micro-vascular damage leading to micro inflammations? Since CRP is present in the atherosclerotic lesion, it may actively contribute to the progression and/or instability of the atherosclerotic plaque. Beyond common risk factors for atherosclerosis, the authors added data on oxidative stress, anti-oxidative capacity and platelet function, which are solely acquired as baseline in the control group.

The authors present an important analysis, aimed to emphasize the need for a prospective comparative hospital-based analysis. But their recruitment of the control group is unfortunately not comprehensible. What is the added value from the control group?

Author Response

The authors focus on the risk of developing carotid atherosclerosis after chemoradiotherapy for nasopharyngeal carcinoma within a prospective case-control study.

  1. It’s a 4-fold risk? This is a highly interesting topic, especially with the young-age incidence of NPC and the possible curative aim of treatment.  Unfortunately, the patient number is limited (n=36) and data were compared to an even smaller ‘age-matched’ control group (n=22). The demographic data in table 1 describe contrarily a non-balanced comparison. 39% of NPC patients had hypertension, 31% had diabetes and 50% were smokers, while only one risk factor (smoker) was seen in 5% of the control group. 13,8% of NPC patients suffered already from carotid plaque before CCRT. Considering the term of having an age-matched control, those with NPC were in the mean 4 years older. Why only 22 people within the control group?

Answers: Thanks for your comments. We agree with your comment and increase the case numbers of the control group with age and sex-match from the normal reference of our neurosonography laboratory. Finally, we found the annual increase mean CCA IMT (mm) in control and NPC groups were 0.014±0.013 and 0.053±0.025, respectively. We found still a nearly 4-fold increasement after we add the other 14 healthy control in the control group. Although several known risk factors for developing atherosclerosis, all patients receive high radiation doses delivered to either side of CCA. Besides traditional vascular risk factors, we consider radiotherapy remains the major risk factor with respect to vascular risk factors. It is not surprising that radiotherapy is associated with an accelerated risk of carotid atherosclerosis. We agree with your concern, we tone down the statement about a 4-fold increasement after radiotherapy. We deleted “with approximately 4 times more increased” and change it into “with significantly increased” throughout the manuscript.

  1. Why not a matched-pairs analysis (1:1, 1:2) based on the known risk factors for developing atherosclerosis?

Answers: Thanks for your comments. We increased the case numbers of the control group with age and sex-match from the normal reference of our neurosonography laboratory. Besides age, vascular risk factors such as hypertension, diabetes mellitus, and hyperlipidemia are also contributors to the increasement in carotid IMT. Since the presence of hypertension, diabetes mellitus and hyperlipidemia are categorical variables. The severity of vascular risk factors (e.g., hypertension, diabetes mellitus, and hyperlipidemia) cannot be well presented by categorical variables. We add the baseline systolic and diastolic pressure during the ECCS examination and try to make it more clear. The systolic and diastolic blood pressure, HbA1c, cholesterol, LDL, and triglyceride in the NPC group were well-control (Tables 1 and 2). We delete the categorial description of vascular risk factors in Table 1 and try to reflect the severity of vascular risk factors. Although it is more perfect if we enrolled disease control-match volunteers rather than healthy control, it is not reasonable that those disease-control patients have the IMT and laboratory data at both baseline and 9-years follow-up in the real world. Finally, we enrolled healthy control from the reference of our neurosonography laboratory. We hope you can understand.    

  1. One NPC patient had carotid stenting. Vascular risk factors? We should be really aware of a higher incidence of CCA atherosclerosis in NPC patients after radiotherapy and clearly need more data for risk assessment. Atherosclerosis is a multifactorial disease, mainly driven by inflammatory reactions. It’s the same for radiation-induced atherosclerosis, or predominantly micro-vascular damage leading to micro inflammations? Since CRP is present in the atherosclerotic lesion, it may actively contribute to the progression and/or instability of the atherosclerotic plaque. Beyond common risk factors for atherosclerosis, the authors added data on oxidative stress, antioxidative capacity, and platelet function, which are solely acquired as the baseline in the control group.

Answers: Thanks for your comments. Besides radiotherapy, the only patient, who received carotid stenting for critical carotid stenosis, is a heavy smoker but without other common vascular risk factors (e.g., hypertension, diabetes, and hyperlipidemia). Although one meta-analysis showed a nonlinear dose-response relationship between the number of cigarettes smoked per day and the risk of cardiovascular events, the other study showed smoking is associated with the dose-related increase of carotid IMT

Furthermore, this study proceeds and follow-up ECCS in the NPC group for nearly one decade. The healthy control group only receive an annual ECCS study but biomarkers for oxidative stress, antioxidative capacity, and platelet function were only obtained at baseline. Concerning your concern, we added the following sentence in the limitation of the study. They are as follows:

The only patient, who received carotid stenting for critical carotid stenosis, is a heavy smoker but without other common vascular risk factors (e.g., hypertension, diabetes, and hyperlipidemia). Cigarette smoking is a well-known vascular risk factor for atherosclerosis. Although one meta-analysis showed a nonlinear dose-response relationship between the number of cigarettes smoked per day and the risk of cardiovascular events, the other study showed smoking is associated with the dose-related increase of carotid IMT

This study proceeds and follow-up ECCS in the NPC group for nearly one decade. The healthy control group only receive an annual ECCS study but biomarkers for oxidative stress, antioxidative capacity, and platelet function were only obtained at baseline.

  1. The authors present an important analysis, aimed to emphasize the need for a prospective comparative hospital-based analysis. But their recruitment of the control group is unfortunately not comprehensible. What is the added value from the control group?

Answers: Thanks for your comments. We increased the case numbers of the control group with age and sex-match from the normal reference of our neurosonography laboratory. Besides age, vascular risk factors such as hypertension, diabetes mellitus, and hyperlipidemia are also contributors to the increasement in carotid IMT. Since the presence of hypertension, diabetes mellitus and hyperlipidemia are categorical variables. The severity of vascular risk factors (e.g., hypertension, diabetes mellitus, and hyperlipidemia) cannot be well presented by categorical variables. We add the baseline systolic and diastolic pressure and try to make it more clear. The systolic and diastolic blood pressure, HbA1c, cholesterol, LDL, and triglyceride in our NPC group were well-controlled (Tables 1 and 2). Although it is more perfect if we enrolled disease control-match volunteers rather than healthy control, it is not reasonable that those disease-control patients have the IMT and laboratory data at both baseline and 9-years follow-up in the real world. Finally, we enrolled healthy control from the reference of our neurosonography laboratory. We hope you can understand.     

Reviewer 2 Report

Authors should be congratulated on the study with a long-term follow-up.

Methodology of factors of interest is appropriate. Anyhow, comprehension of radation dose delivered to the site of CCA IMT assessment would add scientific value to the work. We know the extent of late normal tissue damage by ionizing radiation is a dose dependent. As the anatomical point(s) of  interest is known it should be quite easy to read the radation dose from radiotherapy plans. In this way an interesting information, the relationship between IMT change and the mean dose delivered, could be evaluated.

Please, chceck the writing as follows:

261: PNC ... NPC

281 - 283: To date, only little clinical researches and ours... To date only a few clinical researchers...

319 - 320:  Further, our patients who died had a significantly low antioxidant capacity and a trend higher but nonsignificant higher oxidative stress. ... This notion has to be interpreted cautiously. This is a retrospecitve observation of just a single significant difference in one factor (thiol) with doubtful relevance. Consider to ignore this finding.

327: Our study also showed... Our previous study also showed...

341-342: epidemiological studies cannot be really assessed... epidemiological studies cannot really assess...

362- 362: ECCS may be added in annual screenings in NPC patients to early detect radiation‐related extracranial vasculopathy and also to reduce future
cerebrovascular events. ... Consider this recommendation as relationship between radiation‐related extracranial vasculopathy and the risk of  cerebrovascular events has not been firmly established and in case we believe there is some, we do not know the preventive measures i.e. how and when to act.

Author Response

Reviewer 2)

  1. Authors should be congratulated on the study with a long-term follow-up.

Answers: Thanks for your comment

  1. Methodology of factors of interest is appropriate. Anyhow, comprehension of the radiation dose delivered to the site of CCA IMT assessment would add scientific value to the work. We know the extent of late normal tissue damage by ionizing radiation is dose-dependent. As the anatomical point(s) of interest is known it should be quite easy to read the radiation dose from radiotherapy plans. In this way, a piece of interesting information, the relationship between IMT change and the mean dose delivered, could be evaluated.

Answers: Thanks for your comment. We added the mean radiation dose delivered to either side of CCA IMT and assess the relationship between IMT change (the value of IMT at the 9-years follow-up minus the value of IMT at baseline) and the mean dose delivered according to your comment. We found the relationship between IMT change and the mean dose delivered did not show statistical significance. Since all patients receive high radiation doses delivered to either side of CCA, it is not surprising that the correlation analysis did not show statistical significance between IMT change and the mean dose delivered. In contrast, the IMT change showed a significant increase as compared with the control group. We add the sentences in Method, Result, and Discussion sections. They are as follows:

Assessment of the relationship between the radiation dose and the CCA IMT change

The mean radiation dose on right and left CCA were 6769.8±510.1 and 6788.7±488.7 cGy, respectively. The CCA IMT changes are defined as the value of IMT at the 9-years follow-up minus the value of IMT at baseline. Correlation analysis was explored to evaluate the relationship between the mean radiation dose delivered to either side of CCA and the ipsilateral side CCA IMT change. The results showed as follows: right CCA IMT change and mean radiation dose (r=0.38, p=0.15) and left CCA IMT change and mean radiation dose (r=0.33, p=0.21).

Regarding the dose-response relationship, we found the relationship between IMT change and the mean dose delivered did not show statistical significance. Since all patients receive high radiation doses delivered to either side of CCA, it is not surprising that the correlation analysis did not show statistical significance in NPC patients. In contrast, the IMT change showed a significant increase as compared with the control group.

  1. Please, check the writing as follows: 261: PNC ... NPC

Answers: Thanks for your comment. It is our typing mistake. We correct the mistake.

  1. 281 - 283: To date, only little clinical research and ours... To date only a few clinical researchers...

Answers: Thanks for your comment. We change the sentence “To date, only little clinical research and ours” into “To date only a few clinical researchers” according to your comment.

  1. 319 - 320: Further, our patients who died had a significantly low antioxidant capacity and a trend higher but nonsignificant higher oxidative stress. ... This notion has to be interpreted cautiously. This is a retrospective observation of just a single significant difference in one factor (thiol) with doubtful relevance. Consider ignoring this finding.

Answers: Thanks for your comment. We agree with your comment. We deleted the description according to your comment.

  1. 327: Our study also showed... Our previous study also showed...

Answers: Thanks for your comment. We corrected it according to your comment.

  1. 341-342: epidemiological studies cannot be really assessed... epidemiological studies cannot really assess...

Answers: Thanks for your comment. We corrected it according to your comment.

  1. 362- 362: ECCS may be added in annual screenings in NPC patients to early detect radiation‐related extracranial vasculopathy and also to reduce future cerebrovascular events. ... Consider this recommendation as the relationship between radiation‐related extracranial vasculopathy and the risk of cerebrovascular events has not been firmly established and in case we believe there is some, we do not know the preventive measures i.e. how and when to act.

Answers: Thanks for your comment. We added one paragraph in the Discussion section about the management of radiation-related extracranial vasculopathy. They are as follows:

Management of radiation-related extracranial vasculopathy

Therapeutic guidelines for the prevention of cerebral infarction or transient ischemic attack in extracranial carotid disease recommend that optimal medical therapy, including antiplatelet therapy, statin therapy, and risk factor modification while interventional approaches including carotid endarterectomy, carotid angioplasty, and stenting are reasonable to those patients with symptomatic extracranial carotid stenosis (>70%). However, no clinical trial to date has adequately assessed the medical treatment options in primary or secondary stroke prevention in this selected group of patients. The effect of antiplatelet, anticoagulant, antihypertensive, or lipid-lowering therapy in limiting disease progression is therefore unclear and highlights the need for more clinical studies to develop additional drugs.

By the way, we also revised the sentence in the conclusion section and try to make it clear.

ECCS may be added in annual screenings in NPC patients to early detect radiation‐related extracranial vasculopathy really and also to assess the discrepancy between radiation-related extracranial vasculopathy and symptomatic cerebral infarction. Although no clinical trial to date has adequately assessed the medical treatment options in primary or secondary prevention in radiation‐related extracranial vasculopathy, optimal medical therapy, including antiplatelet therapy, statin therapy, and risk factor modification could be effective.